# High Nature Value Farming Systems and Protected Areas: Conservation Opportunities or Land Abandonment? A Study Case in the Madrid Region (Spain)

**María F. Schmitz** [1,*], **Cecilia Arnaiz-Schmitz** [2] **and Patricio Sarmiento-Mateos** [3]

1 Department of Biodiversity, Ecology and Evolution, Universidad Complutense de Madrid, 28040 Madrid, Spain

2 Department of Civil Engineering, Transport, Territory and Urbanism, Higher Technical School of Engineering of Roads, Channels and Ports, Universidad Politécnica de Madrid, 28040 Madrid, Spain; cecilia.arnaiz@upm.es

3 Regional Ministry of Education and Youth of the Government of Madrid, 28014 Madrid, Spain; patricio.sarmiento@educa.madrid.org

* Correspondence: ma296@ucm.es

**Abstract:** European rural landscapes contain high nature value farmlands that, in addition to being the main economic activity in many rural areas, host habitats and species of great conservation value. The maintenance of these farming systems largely depends on traditional ecological knowledge and the rural lifestyles of the local populations. However, they have not been sufficiently appreciated and protected, and as a result, they are currently threatened. In this study, which was performed in the Madrid region (central Spain), we analyse the social-ecological changes of the rural landscape after the establishment of a protected natural area network. The obtained results highlight a significant loss of these high nature value farming systems and a marked increase in the rewilding processes characterised by scrub–forest transition and the development of forest systems. These processes are linked to the disruption of the transmission of traditional ecological knowledge, which may imply negative consequences for both the high biocultural diversity that these systems host and the cultural identity and the socioeconomics of the rural populations that live there. A useful methodological tool is provided for social–ecological land planning and the design of effective management strategies for the conservation of rural cultural landscapes.

**Keywords:** social–ecological systems; cultural rural landscape; protected areas; rewilding; rural socioeconomics; forest expansion; rural to urban land conversion; biocultural heritage; biodiversity; naturalness

## 1. Introduction

The theory of social–ecological systems arose from the recognition of the close interaction between nature and society [1]. Among the different types of systems, rural cultural landscapes are characterized by the conservation and protection of ecological processes, natural resources, and cultural biodiversity [2–4]. Secular interactions between humans and nature have given rise to a wide variety of sustainable cultural landscapes that have primarily been maintained over time with traditional ecological knowledge (TEK). This cultural process is based on the transmission of the deep empirical knowledge of the sustainable use of natural resources and, therefore, of the conservation of ecological processes and biodiversity [5].

In recent decades, significant efforts have been made to preserve TEK and cultural landscapes in Europe [6]. The European countryside is characterized by a rich array of rural cultural landscapes that have been shaped over millennia by traditional land uses [7,8]. The dynamic interrelationship between social and ecological systems has given rise to a broad range of cultural agricultural landscapes that, in addition to their primary functions

of producing food and fibre, are rich in natural and/or semi-natural vegetation and often harbour habitats and species of great conservation value [9,10]. These "high natural value farming systems" (HNVF), with their strong heritage significance and outstanding aesthetic and recreational qualities, favour the conservation of biodiversity and provide society with a great variety of essential ecosystem services that have improved the living standards of local populations and have resulted in valuable cultural landscapes [7,11,12]. The HNVF concept emphasizes the role of certain types of farmlands in the conservation of biodiversity in rural areas [13]. At present, European agricultural biodiversity is considered to be as valuable as wild biodiversity [14].

Despite the indicated characteristics, and probably also due to their everyday use, rural cultural landscapes and their associated HNVF have not been sufficiently valued and protected. This has involved the rapid and radical transformations of traditional land-use systems across Europe in recent decades, mainly as a consequence of the continuous process of polarization derived from the main land use change trajectories: extensification and rural abandonment, on the one hand, and intensification and urban expansion, on the other [15–19]. The rate and extent of technological, economic, and cultural changes threaten cultural landscapes and the rural societies associated with them [7,16,20].

In this context, protected areas (PAs) could play a key role in the protection and maintenance of European HNVF because they are social–ecological systems whose sustainability and management are strongly influenced by people [21]. PAs are central to conservation strategies, but the effectiveness of some measures included in their regulatory schemes can be questioned [22]. Therefore, several studies highlight that PAs often restrict rural activities and neglect local populations with respect to their TEK, their historical and cultural context, and their important contribution to the maintenance of these cultural landscapes. These restrictions promote the abandonment of agricultural land and traditional management practices, causing the loss of biocultural diversity [20,23–26]. Thus, nature conservation strategies supported by PAs have often promoted the abandonment of farmlands, pastures, and cultural forests that host high biodiversity and that are being transformed into mosaics of scrub and mixed forests and forest systems. These processes of rewilding (returning ecosystems to a higher level of naturalness, seeking wildlife comeback without human intervention), cause the disappearance of HNVF, fostering spatial homogeneity and the degradation of the cultural landscape [7,25,27]. This restrictive approach to conservation favours "inside-out" processes, generating the development of different opportunities inside and outside the limits of the parks [20,26,28–31]. So far, no land planning and management schemes have been achieved that can provide effective designs and responses to safeguard the values of cultural landscapes and traditional land-use systems, which are still at risk today. Therefore, the future of HNVF is very uncertain [17,32,33].

The overall objective of this paper is contextualized within the conceptual framework of social–ecological systems and, specifically, within the study of the effectiveness of PA guidelines and management in relation to the protection and maintenance of HNVF. The baseline for this study is the research carried out by Sarmiento-Mateos et al. (2019) [22], which, from a scientific perspective, focused on the evaluation of normative documents and the guidelines for the planning and management of two Spanish PAs under different protection. The findings from that research highlight various weaknesses and inconsistencies in the zoning design and regulation schemes of the studied PAs, which mainly promote uses and activities more related to the nature of the areas than to the culture, causing negative consequences for the cultural landscape that, contradictorily, these legal instruments claim to protect (Figure 1). On this basis, the specific objectives of our research are are: (i) to find out through empirical evidence based on the social–ecological evolution of the study area whether rural land planning and nature conservation strategies by means of the establishment of PAs consider traditional agricultural systems as valuable components of rural cultural landscapes; (ii) to detect inside-out processes related to the dynamics of HNVF in territories with established PAs (i.e., restrictions on agricultural uses and practices within the boundaries of PAs and opportunities outside them, or vice versa);

and (iii) to identify the social and economic impacts of PAs on rural populations living inside of and outside of their boundaries.

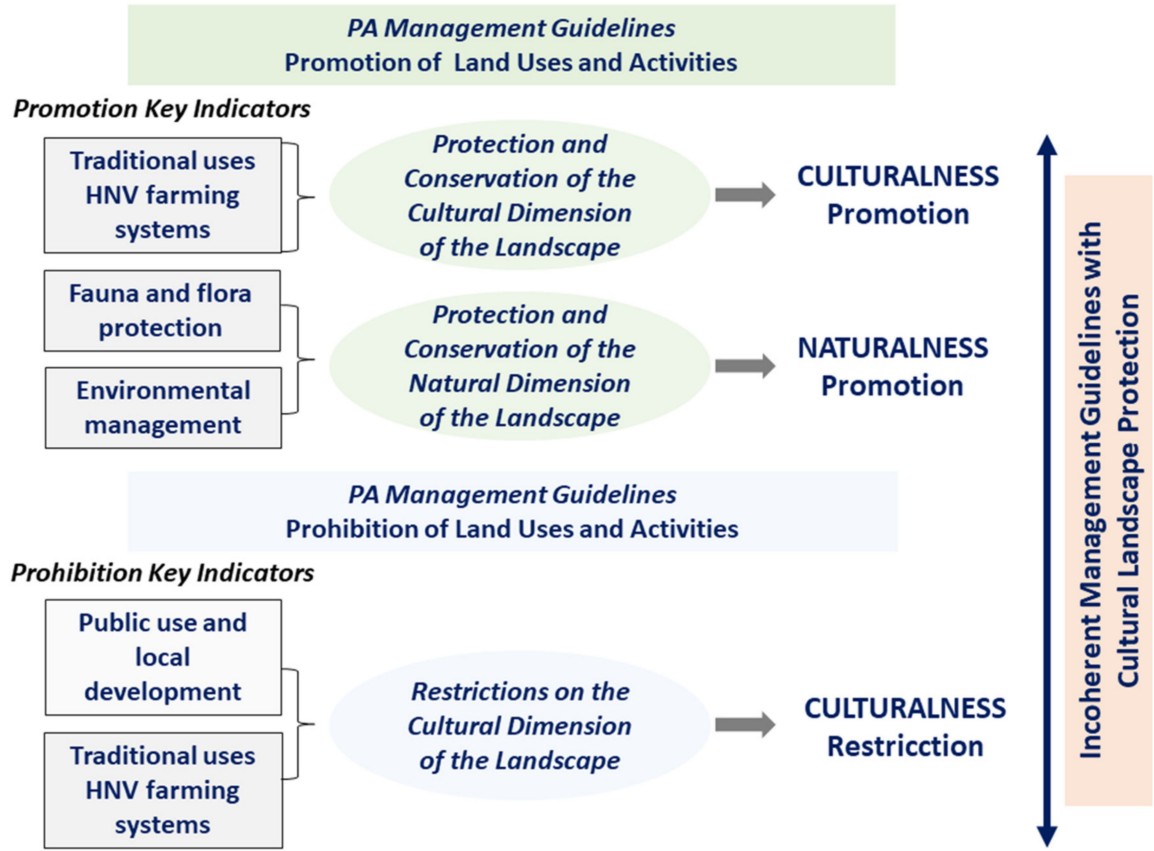

**Figure 1.** General outline of the main indicators of the PA management guidelines, identified from the analysis of the corresponding normative documents. Modified from Sarmiento-Mateos et al. 2019 [22].

## 2. Materials and Methods

### 2.1. Study Area

The study area is located in the north-northwest area of the Madrid region (central Spain) and covers 38 municipalities. As a result of the recognition of the natural and cultural values of this territory, a few decades ago, a PA network began to establish and expand PAs in this area, which consist of the "Cuenca Alta del Manzanares Regional Park" (52,800 ha; declared in 1985), the "Cumbre, Circo and Lagunas de Peñalara Natural Park" (15,030 ha; declared in 1990) and, later, the "Sierra de Guadarrama National Park" (33,960 ha, of which 21,714 ha are in the Madrid region; declared in 2013). After the declaration of the National Park, the limits of this space partially overlapped with not only thousands of hectares of the Regional Park but also overlapped practically the entire area belonging to the Natural Park, which became integrated into the former. As a consequence of the establishment of the national park, the natural park was derogated as a protection category of the territory (Figure 2).

Because of its location, is the study area is of a continental Mediterranean climate, characterised by hot, dry summers and cold, wet winters (according with the Köppen classification). In the study area, the average annual temperature ranges from 4 °C in the summits of the mountains to 13.5 °C in the foothills. Average rainfall ranges from 450 mm to 1615 mm per year. The substrate is formed by granitoid and gneiss rocks with lithic and dystric leptosols. The main environmental characteristic is the marked altitudinal variation, which is between 600 and 2383 m a.s.l. and favours the development of different vegetation belts. The natural vegetation corresponds to Mediterranean forests, with different species

of trees such as Quercus ilex, Q. pyrenaica, Q. faginea, and Fraxinus angustifolia as well as scrubs such as Cistus ladanifer, Cytisus scoparius, Lavandula stoechas, and Genista cinerea, among others.

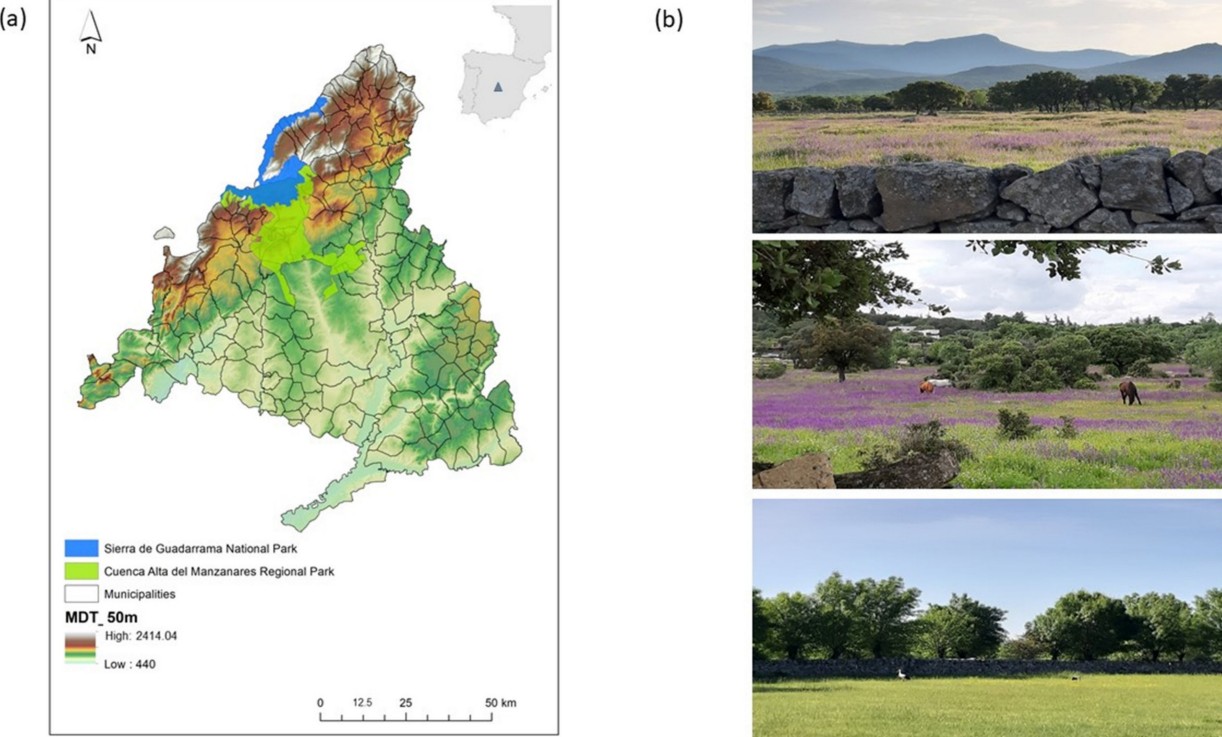

**Figure 2.** (**a**) Location of the study area in the Madrid region (central Spain). The two main categories of landscape protection at present (Regional Park and National Park) are shown, as well as the overlapping area between both parks; (**b**) HNVF characteristics of the study area (mainly silvopastoral systems).

The area is an ancestral mountainous cultural landscape that is mainly composed of traditional agrosilvopastoral land uses with relevant HNVF, mainly pastures and "dehesas" at low and medium altitudes immersed in a bocage type rural matrix with drove roads, hedgerow networks, woodland, and enclosures of stone walls as unique and characteristic elements of the landscape [20,26]. Dehesas (open savannah-like woodlands used as pastures) are human-made systems that combine exploitation with habitat conservation and support high habitat heterogeneity and great natural and social values while providing relevant ecosystem services and biodiversity conservation [34,35]. For these reasons, dehesa systems are considered a model for sustainable agriculture, and their conservation depends on the maintenance and effective management of traditional land uses [25,36]. Highlands have traditionally been used as summer pastures for native livestock breeds, such as the Avileña-Black Iberian cattle, which moves seasonally from the pasture systems of the valleys to the upland grasslands (this altitudinal movement of livestock is called trasterminance) through a wide network of drove roads of great cultural value [25,37,38]. The result of this complex social–ecological relationship is a multifunctional rural landscape that provides numerous provisioning and regulating ecosystem services and high biocultural diversity [20,39].

For centuries, these HNVF have been the main economic activity for local populations and have constituted a major factor in the shaping of the current landscape [22]. Recognition of the value of this multifunctional landscape and its accelerated dynamics of change, similar to many other European rural landscapes, has led to the design and application of nature conservation strategies through the establishment of different categories of PAs in the last 40 years. Regional and Natural Parks combine the protection and use of sustainable

landscapes. Thus, among the objectives contained in their regulatory frameworks is the promotion of the maintenance, recovery, and implantation of the traditional productive activities of an agricultural, livestock, and forestry area as a means to ensure the survival of natural and cultural values [40,41]. Despite this, the evolution of the territory after the establishment of the first PAs has favoured the raising of the protection regime to that of a National Park, the highest level of protection allowed by the Spanish legal system. Its main objective is to ensure the conservation of the natural values of the protected territory. Spanish National Parks are places where "non-intervention" prevails and where the principle is to allow the free evolution of natural processes [42].

### 2.2. Selection of Land Analysis Units

This paper focuses on the study of the social-ecological evolution experienced by the studied rural cultural landscape, the relevant HNVF's of which were the main reason for the establishment of a PA network several decades ago. The spatial-temporal analysis units were both the municipalities included within the limits of the PA network and the ones closest to them in the influenced area surrounding the territory of the parks. We analysed a total of 38 municipalities, 15 of them totally or partially included within the PA network (with more than 25% of the municipal area) and 23 municipalities outside the network (with a municipal area < 25% or not included within its boundaries) (Appendix A).

Municipalities are an interesting and effective local scale of analysis, and several authors encourage their use in landscape studies [43–45]. In Spain, municipalities are the smallest administrative units of land management and the most detailed scale of decision-making. Furthermore, socioeconomic information is recorded at this level [19,20,46].

### 2.3. Data Collection

We registered quantitative descriptors of the social-ecological variation of the studied territory. These descriptors can be considered representative of HNVF and the socioe-conomics of the local populations. Thus, for each municipality, we selected (a) a set of 10 descriptors of the HNVF in the study area that were linked to farmland dominated by low intensity farming practices and whose nature value results come primarily from (i) a high proportion of semi-natural vegetation; (ii) a mosaic of semi-natural and cultivated land; and (iii) a high diversity of land use–land cover (LULC) combined with semi-natural elements [47,48]. These HNVF descriptors were obtained from the reclassification of 27 LULC from pre-existing land use maps (SIGA public database, 1990–2010; Table 1a); (b) 11 socioeconomic descriptors (Table 1b) related to local population structure (population density, population aging degree), population dynamics (emigration), economic living conditions (income per capita), labour market (unemployment in the agricultural sector, agricultural workers), contribution of gross domestic product (GDP) to the local economy (agricultural GDP, industrial GDP and GDP from service sector), and characteristics of the land planning (urban land area, agricultural land area). Socioeconomic data were obtained from regional public censuses [49].

Data of the selected social-ecological descriptors (LULC and socioeconomic variables) were recorded in two periods, 1990–2000 and 2000–2010, prior to the declaration and establishment of the National Park in the study area. With the collected social and economic data, we elaborated four quantitative matrices, describing the 38 municipalities in the two time periods by means of the 10 descriptors based on the most representative land uses of HVN farming systems and 11 descriptors of the socioeconomic structure of local populations, respectively. LULC were quantified as the occupied area in relation to the municipal area (Table 1a). Socioeconomic data were recorded at the municipal level and their units of measurement varied depending on the type of descriptor used (see Table 1b). Data recorded in the matrices were the average values of each social-ecological descriptor in each period.

**Table 1.** Social-ecological descriptors recorded in each municipality of the study area. A brief description and units of measurement are indicated. (a) Land use–land cover descriptors considered as representative of high nature value farmlands of the studied cultural landscape. The unit of measurement was percentage area in relation to municipal area; (b) socioeconomic variables related to local population structure.

| (a) | |
|---|---|
| **Land Use Variables** | **Description** |
| Pastures | Pastureland; scrub-pastureland mosaics |
| Dehesas | Open formations with a mixture of pastures and isolated trees. Pastures with broadleaf tree species; pastures with coniferous species; pastures with mosaics of scrub and tree species (broadleaf trees and/or coniferous) |
| Herbaceous crops | Irrigated herbaceous crops; orchards and forced crops; rainfed herbaceous crops; mosaics of crops and broadleaf tree species |
| Woody crops | Rainfed olive groves; rainfed vineyard; rainfed mixed crops of olive grove and vineyard; irrigated fruit tree crops; mosaics of crops of fruit trees with conifers |
| Semi-natural meadows | Semi-natural meadows; mosaics of semi-natural meadows with broadleaf tree species |
| Shrubland | Mediterranean shrubland dominated by high cover of scrubs of different types and multiple uses (grazing of domestic and wild herbivores, honeybee colonies . . . ) |
| Systems in scrub-forest transition | Shrubby and woody vegetation. Associations of scrub-coniferous, scrub-broadleaf tree species, scrub-coniferous and broadleaf tree species |
| Mixed forests | Multi-specific and heterogeneous forests |
| Broad-leaved forests | Mediterranean broad-leaved sclerophyllous and deciduous forests. Forests of Holm oak (*Quercus ilex*), Pyrenee oak (*Q. pyrenaica*), juniper (*Juniperus oxycedrus*), and different types of scrubs |
| Coniferous formations | Montane pine forests and plantations of Scots pine (*Pinus sylvestris*) and black pine (*P. nigra*) in hillside slopes and pinaster pine (*P. pinaster*) in the valley bottoms |
| (b) | | |
| **Socioeconomic Variables** | **Description** | **Units** |
| Population density | Number of inhabitants per $km^2$ | Inhabitants/$km^2$ |
| Population aging | Population of 65 years and over in relation to total population | Percentage |
| Emigration | Number of people that have changed their home outside the municipality in relation to the total population | Percentage |
| Income per capita | Disposable income per capita | Euros |
| Agricultural workers | Number of people working in the agricultural sector in relation to the economically active population | Percentage |
| Agricultural unemployment | Number of unemployed in the agricultural sector in relation to the total of registered unemployed | Percentage |
| Agricultural sector GDP | Gross domestic product from the agricultural sector | Percentage |
| Industrial sector GDP | Gross domestic product from the industrial sector | Percentage |
| Service sector GDP | Gross domestic product from the service sector | Percentage |
| Urban land area | Municipal area that has all urban services | Percentage |
| Agricultural land area | Municipal area dedicated to farmland and pastures | Percentage |

*2.4. Data Analyses*

In order to detect and quantify the main social-ecological characteristics of the studied rural landscape and its variation over time, four principal component analyses (PCA) were performed: one for each matrix corresponding to the different land uses and socioeconomic conditions of the municipalities in the different considered time periods. The factor loading of the initial sets of descriptors in the main axes of the PCAs allowed us to identify the main characteristics (tendencies of variation) and indicators of the studied rural landscape as well as their changes over time, both inside of and outside of the limits of the PAs.

To determine the statistical significance of the magnitude of the changes to the LULC and the considered socioeconomic descriptors over time, Student's *t*-tests on two paired samples were used (two series of quantitative measures on the same units) at regional scale and inside of and outside of the PAs.

## 3. Results

*3.1. High Nature Value Farmlands and Landscape Dynamic*

PCAs carried out from the data matrices of the municipalities x HNVF descriptors indicate that in the studied periods, the two main ordination axes obtained express the same landscape variation tendencies. Since the explained variance in the first dimension of the PCA is the highest, only the variation expressed by PCA-axis 1 has been considered in both cases. Figure 3a,b, show the distribution of the municipalities along the first ordination axis in the two considered timeframes. The analyses highlight that the structure of the landscape inside of and outside of the boundaries of the PAs established in the study area and its variation over time are very similar.

**Table 2.** Factor loadings of the land use descriptors on PCAs-Axes 1 of the analysed two periods (variance absorptions are shown in brackets). Loadings of the variables identified as key indicators of the cultural landscape over time are indicated in italics (see Figure 3).

| Land Use Descriptors | PCA-Axis 1 1990–2000 (33.33%) Factor Loadings | PCA-Axis 1 2000–2010 (33.42%) Factor Loadings |
|---|---|---|
| Pastures | −0.531 | −0.268 |
| Dehesas | −0.286 | −0.248 |
| Herbaceous crops | −0.243 | 0.002 |
| Woody crops | −0.166 | 0.115 |
| Semi-natural meadows | 0.767 | 0.591 |
| Shrubland | 0.650 | 0.503 |
| Systems in scrub-forest transition | −0.325 | −0.665 |
| Mixed forests | −0.420 | −0.776 |
| Broad-leaved forests | 0.306 | 0.349 |
| Coniferous formations | 0.705 | 0.653 |

According to the loading of the analysed variables (Table 2; Figure 3a), we can observe that in the first period, the traditional silvopastoral systems that prevail in this territory are only represented by pastures (highest factor loading at the negative end of the PCA-axis 1: −0.531). Similarly, mixed forests (−0.42) and uses linked to the scrub–forest transition (−0.325) are also associated with the negative end of this axis, indicating the great importance that processes such as shrub encroachment and forest expansion have acquired. The positive end of PCA-axis 1 corresponding to the period of 1990–2000 presents land uses characteristic of forest systems with a high degree of naturalness as landscape indicators (in order of importance, according to their weights in the axis formation: 0.767, semi-natural meadows; 0.705, coniferous formations; 0.650, shrubland; 0.306, broad-leaved forests).

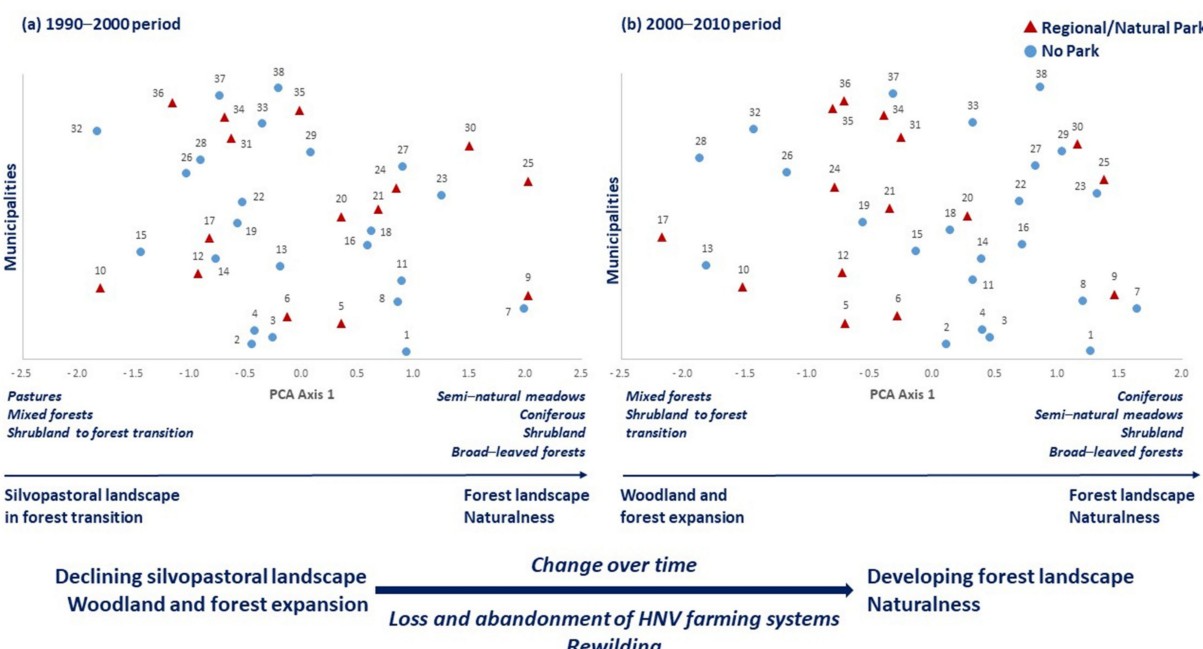

**Figure 3.** Land use dynamics. Coordinates of the municipalities of the study area along PCAs-axes 1: (**a**) period 1990–2000; (**b**) period 2000–2010. Land use indicators (variables with higher loadings in the PCAs) are shown at both ends of the axes (see Table 2). The codes of the municipalities are indicated in Appendix A.

The PCA calculated for the second period (2000–2010) also shows the interspersion between the municipalities of the study area, regardless of their location inside of or outside of the PA network (Figure 3b). The factor loadings of the indicator variables (Table 2) highlight the transformation of the traditional rural landscape structure towards an eminently forested and naturalized landscape. Thus, at the negative end of the axis, the variables with the greatest loading are mixed forests (−0.776) and systems in transition from scrub to forest (−0.665), while the positive end is characterized, in order of importance, by coniferous formations (0.653), semi-natural meadows (0.591), shrubland (0.503), and broad-leaved forests (0.349). The variation tendencies identified from the analyses indicate a rural landscape structure regression process, in which livestock systems have lost importance in favour of woodland and forest landscapes. This process implies a significant loss of HNVF. The degradation of traditional rural landscapes and their associated HNVF has occurred throughout the territory, both inside of and outside of PAs.

Figure 4 represents a significant decrease over time in the area occupied by the main silvopastoral land uses (pastures and dehesas) both on a regional scale (the entire study area; Figure 4a) and local scale (municipalities inside of and outside of the boundaries of the PA network; Figure 4b,c, respectively). At all scales, this abandonment trend and loss of high natural value pasture systems corresponds to the significant increase of the area occupied by forest systems with a lesser degree of human intervention (Figure 3).

### 3.2. Socioeconomic Dimensions of the Rural Landscape

Figure 5 shows the main socioeconomic characteristics of the territory and their variation over the study time. According to the loadings of the of the PCA variables performed on the data matrix corresponding to the 1990–2000 period (Table 3), the first variation tendency represented by PCA-axis 1 highlights a rural–urban socioeconomic gradient (Figure 5a). Its key indicators are the degree of population aging (factor loading: 0.929), the number of workers in the agricultural sector (0.892), and the agricultural GDP (0.875) at the positive end of the axis, and the income per capita of local people (−0.857), the urban land area of the municipalities (−0.753), and their population density (−0.625) at the negative end. The coordinates of the municipalities along PCA-axis 1 indicate similar

characteristics between them, regardless of whether they belong to the PA network or not. However, it is noteworthy that the municipalities with the highest degree of rurality, according to their position on the axis, are not within the limits of the PA.

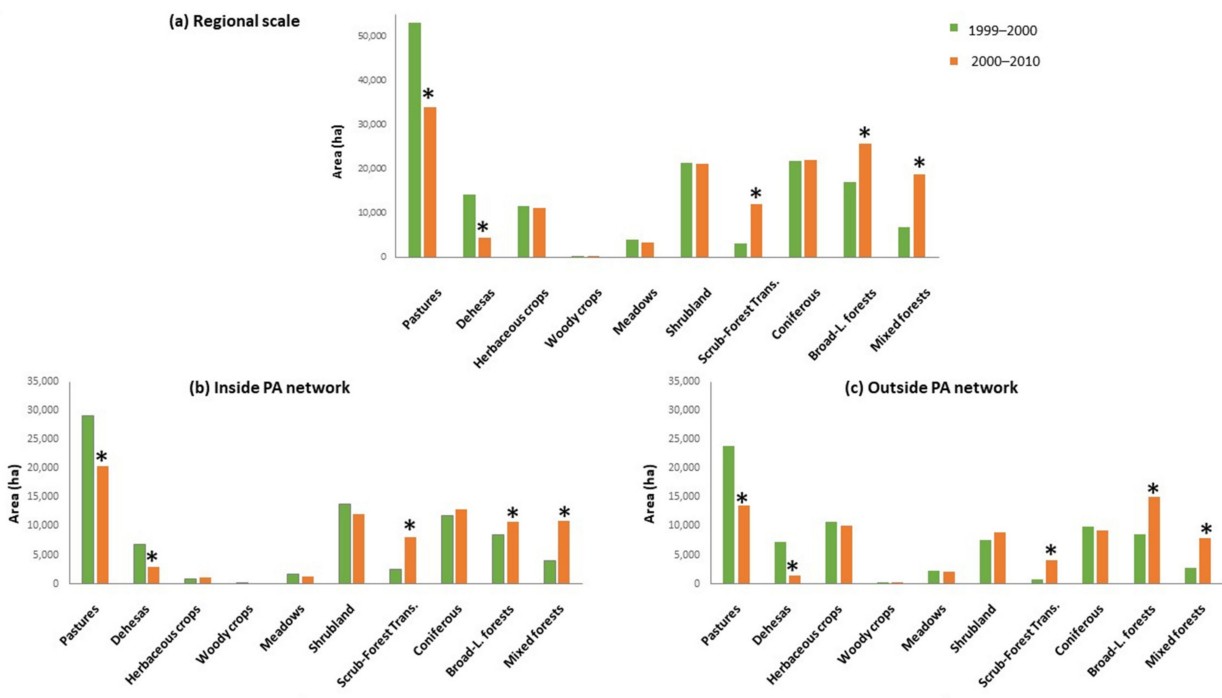

**Figure 4.** Temporal evolution of the land use descriptors of HNVF in the PA network and their surrounding area: (**a**) at a regional scale (study area); (**b**) inside of and (**c**) outside of the PA boundaries. Statistically significant changes (*p* < 0.05) analysed by means of Student's *t* test are indicated with an asterisk.

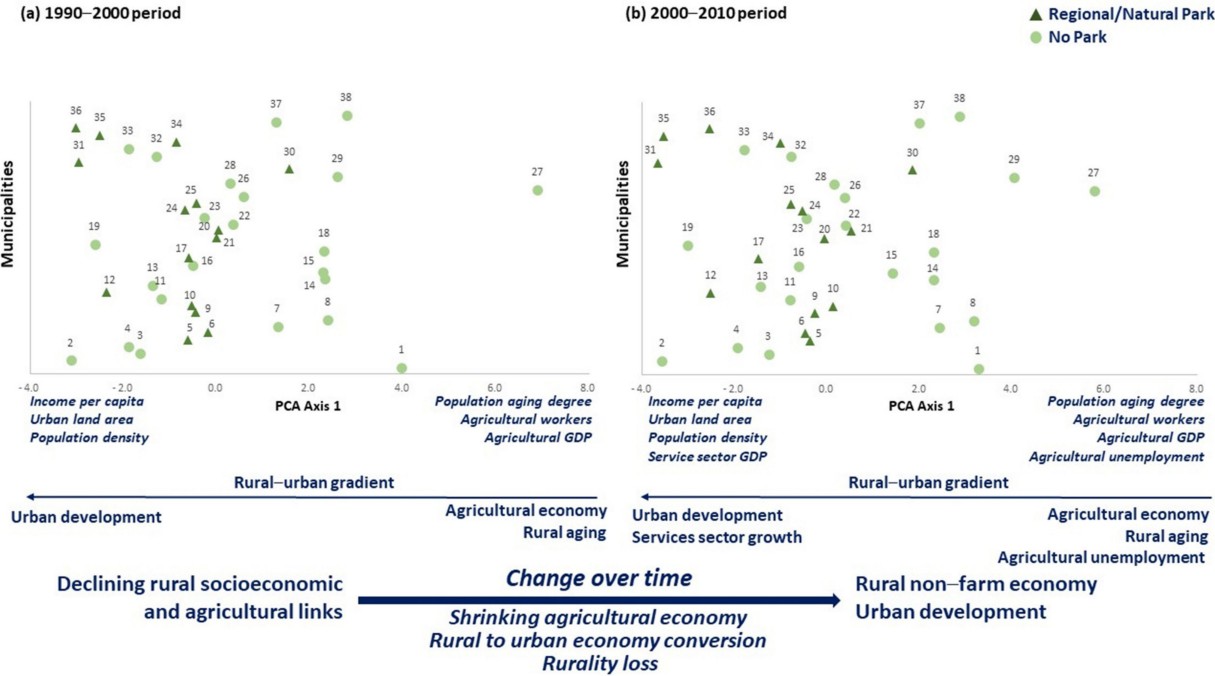

**Figure 5.** Socioeconomic dynamics. Coordinates of the municipalities of the study area along PCAs-axes 1: (**a**) period 1990–2000; (**b**) period 2000–2010. Socioeconomic indicators (variables with higher loadings in the PCAs) are shown at both ends of the axes (see Table 2). The codes of the municipalities are indicated in Appendix A.

**Table 3.** Factor loadings of the socioeconomic descriptors on PCAs-Axes 1 of the two analysed periods (variance absorptions are shown in brackets). Loadings of the variables identified as key indicators of the socioeconomic structure of local population over time are indicated in italics (see Figure 5).

| Socioeconomic Descriptors | PCA-Axis 1 1990–2000 (40.60%) Factor Loadings | PCA-Axis 1 2000–2010 (43.61%) Factor Loadings |
|---|---|---|
| Population density | −0.625 | −0.717 |
| Population aging | 0.929 | 0.843 |
| Emigration | 0.383 | −0.198 |
| Income per capita | −0.857 | −0.885 |
| Agricultural workers | 0.892 | 0.818 |
| Agricultural unemployment | 0.042 | 0.687 |
| Agricultural sector GDP | 0.875 | 0.823 |
| Industrial sector GDP | −0.226 | 0.211 |
| Service sector GDP | −0.265 | −0.507 |
| Urban land area | −0.753 | −0.782 |
| Agricultural land area | −0.286 | 0.136 |

The change of this territory over time accentuates the identified tendency. Thus, both the main socioeconomic indicators detected by the PCA analysis on the 2000–2010 data matrix and the distribution of the municipalities along axis 1 are very similar to those identified in the previous period. The main difference between both time lapses is the development of the service sector in the 2000–2010 period (factor loading: −0.507), which is associated with urban expansion on the negative end of the axis, and the unemployment in the agricultural sector (0.687), which is linked to rural economy on its positive end. Those municipalities that belong to the PA network are preferably related to urban development conditions (Figure 5b). From these analyses, a marked trend of change can be observed from both inside of and outside of the boundaries of the PA network, with a declining rural socioeconomic link to agrarian systems with movement towards a new rural non-farm economy that is related to urban development and a growing economic service sector as well as the decrease of traditional rural activities (Figure 5).

The statistical significance of the changes in the key indicators of the socioeconomic structure obtained from PCA analyses are indicated in Figure 6. It shows the significant increase in urban land area and the decrease in agricultural GDP at both the regional scale and inside of and outside of the PA network (Figure 6a–c). Other statistically significant socioeconomic indicators are the number of workers dedicated to the agricultural sector throughout the territory, the influence area of the parks (Figure 6a,c respectively), and the increase in agricultural unemployment in the municipalities located inside the boundaries of the PA network (Figure 6b).

*3.3. National Park Establishment. Social-Ecological Conditions of a Changing Protected Landscape*

Figure 7 highlights the social-ecological characteristics of the territory that was proposed and later declared as a National Park and its surrounding area. The 2013 declaration that a portion of the studied territory be deemed a National Park involved 14 municipalities, which were considered those with more than 25% of the municipal area inside the boundaries of the park (Appendix A). At the time of the establishment of the National Park, eight of the municipalities belonged to the previous PA network (Regional or Natural Park) (Figure 7a, dark blue triangles within the light blue-shaded area), while the remaining six municipalities were unprotected (Figure 7a, dark blue circles within the light blue-shaded area). Based on the previously performed analyses (see Figure 3), we can verify how the territory prior to the establishment of the National Park was characterized by both the decline of silvopastoral systems and the expansion of woodland and forest as well as the promotion of naturalness. The selection of the area as a new National Park (Figure 7a, blue-shaded area) has prioritized the protection of forest landscapes over the conservation

of silvopastoral landscapes of high natural and cultural value, and to a great extent, they have been left outside of the National Park (Figure 7a gray triangles and circles at the negative end of PCA Axis 1, outside the blue-shaded area).

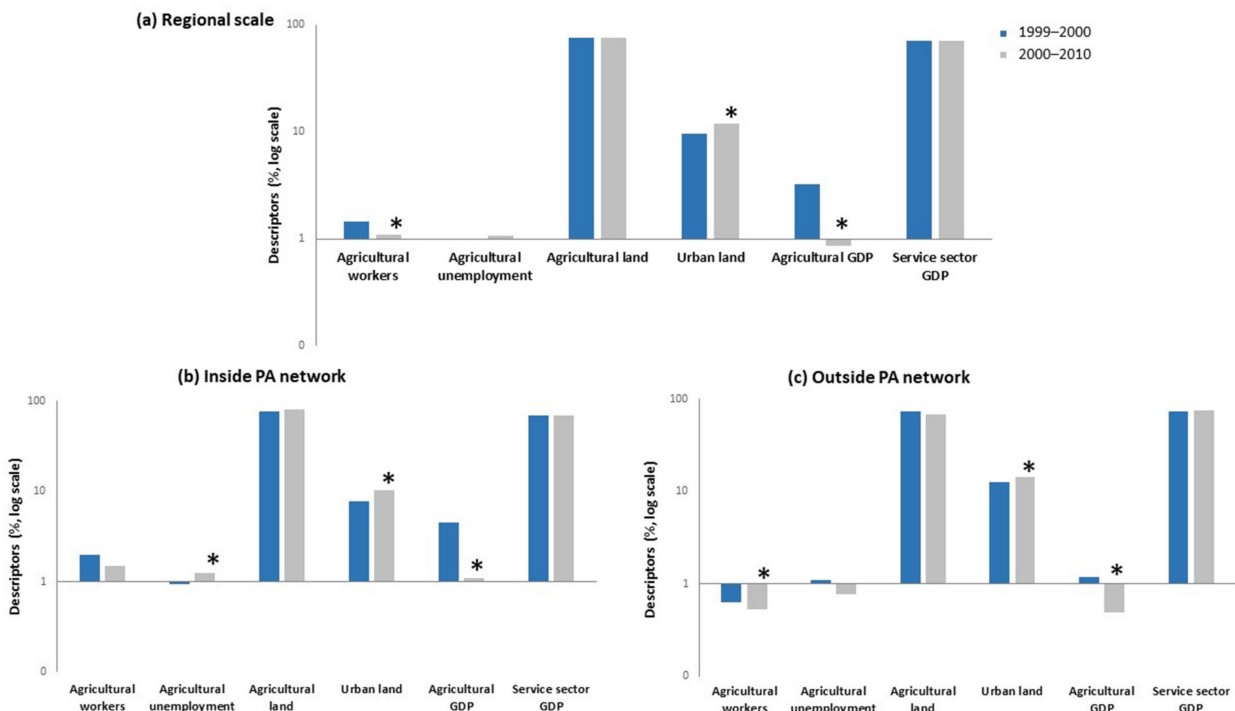

**Figure 6.** Temporal evolution of the socioeconomic descriptors of the local populations linked to HNVF (**a**) at a regional scale (study area); (**b**) inside of, and (**c**) outside of the PA boundaries. Statistically significant changes ($p < 0.05$) analysed by means of Student's *t* test are indicated with an asterisk.

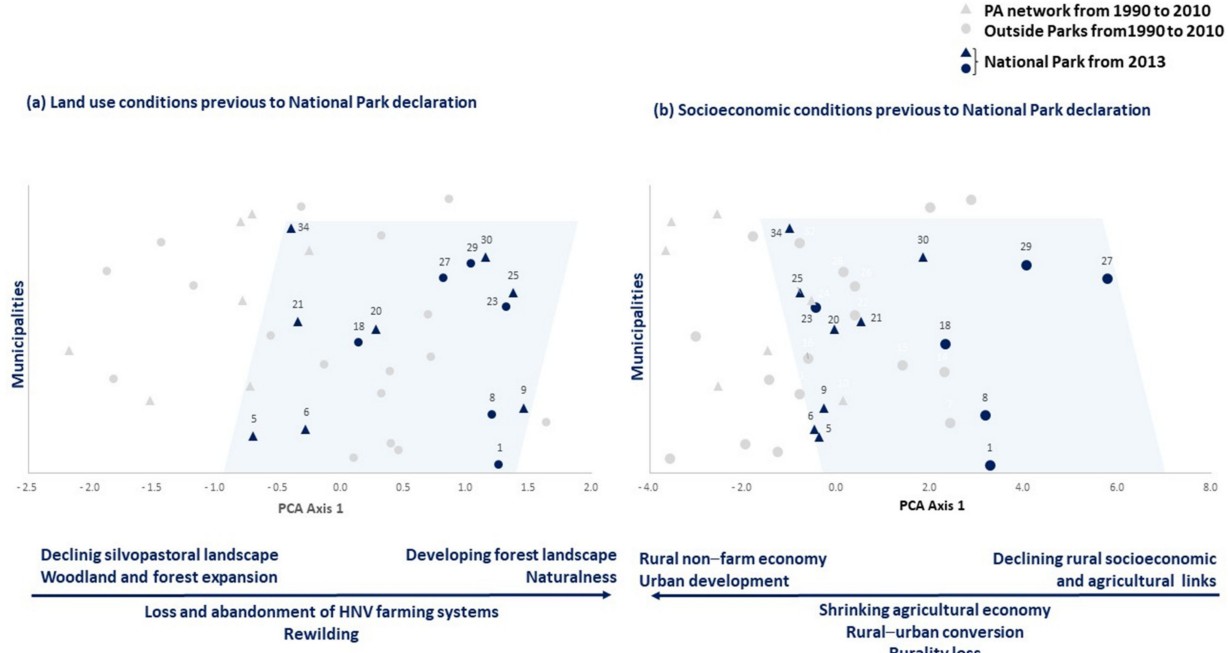

**Figure 7.** Social-ecological characteristics of the study territory prior to the establishment of the National Park (from the results represented in Figures 3 and 5). The blue-shaded area highlights the municipalities included inside the boundaries of the National Park. (**a**) Land use conditions; (**b**) socioeconomic conditions.

Regarding the socio-economic structure of the territory belonging to the National Park, protection has focused both on some municipalities with a high degree of rurality as well as on others characterized by a considerable level of urban development and the transition from an agriculture-based economy to a service-based one (see Figures 5 and 7b, blue shaded area).

From a social-ecological perspective, the obtained results reveal a certain inconsistency in the selection of the municipalities that would belong to the future National Park. In many cases there are no obvious differences (ecological and socioeconomic) between the selected municipalities and those that would remain outside of the park.

## 4. Discussion

The cultural character of European rural landscapes has given rise to a rich heritage built over centuries and is representative of both the historical interaction and co-evolution of the natural and social systems and the biocultural diversity of these landscapes [7]. These anthropogenic landscapes often have a high conservation value that depends on the maintenance of traditional agricultural systems and practices [33]. In this paper, the obtained results underline the trend of abandonment and degradation of the studied cultural landscape, which is immersed in an evident gradient of socioeconomic transition from traditional rural conditions to urban ones and is largely responsible for the pressure to which HVN farming systems are currently subjected. The establishment of a conspicuous PA network has not prevented this transformation process of the rural landscape, but rather, it seems to have accentuated it. Numerous studies emphasize that the lack of adequate and effective land planning and management aimed at the conservation of cultural landscapes and their associated HNVF together with institutional deficiencies in supporting local populations and their own TEK have favoured the abandonment of rural landscapes and traditional land uses and have placed the HNVF in a vulnerable position ([17,20,25], among others).

This study, carried out in a county of the Madrid region, with a wide PA network that has been established for years, was performed from a social-ecological approach using an easily replicable methodological development, which has allowed us to quantify the evolution of the territory and the degree of affectation of the HNVF over the last few decades since the declaration of the different PA categories. The design inside of and outside of the PAs, considering municipalities as units of analysis, has been effective in achieving the proposed objectives. In the area, we have detected a similar tendency of LULC change both inside of and outside of PAs (Figure 3). Throughout the studied period, there has been an important and statistically significant decrease in traditional silvopastoral uses (pastures and dehesas) as well as a notable rewilding process, with an evident increase in woodlands and systems in forest transition (Figures 3 and 4). This entails the change of the cultural landscape towards naturalness and the abandonment and loss of HNVF, regardless of their status or category of protection. Land abandonment and passive rewilding results in the degradation of the rural cultural landscape and the interruption of the TEK transmission, which is essential for its maintenance [50,51]. In Europe, the rewilding of cultural landscapes will be one of the most important landscape changes in the coming decades [52].

The changing landscape that has been described is driven by social and economic factors, but it also is dependent on current environmental policies and the lack of support for the rural population, both politically and economically, which hinders the profitability of traditional agricultural practices [17,53,54]. Consequently, forests are increasing in areas that have been abandoned by livestock. These afforestation and rewilding processes that are currently underway are probably induced and supported by the idea of human-caused environmental transformation and degradation and the need to recover a "natural state" [7,25]. The concepts of naturalness and wilderness have been widely used as a point of reference for the conservation, restoration, and management of ecosystems, especially in the nature conservation strategies promoted by the PA guidelines [22]. However, in

cultural landscapes with a long history of interaction with humans, such as the one studied here, it seems incongruous to take naturalness as a reference to design conservation plans. The main arguments against rewilding include the loss of valued cultural landscapes, a decrease in landscape heterogeneity, negative impacts on biodiversity and ecosystem services, and an increase in human–wildlife conflicts [55–57]. In this regard, segregation between nature and culture, called "cultural severance", has been described as a serious problem in the conservation of natural and cultural heritage [58].

The socioeconomic conditions of local populations have shaped cultural landscapes and their associated HNVF over the previous centuries (Arnaiz-Schmitz et al., 2018b). Changes in the local socioeconomic system are the major driving forces of changes in land use, as landscape and socioeconomic components constitute a co-evolving system [59,60]. In this case study, a range of socioeconomic factors have allowed us to identify a marked process of urban–rural transition maintained over time, both inside of and outside of the boundaries of the PA network (Figure 5). The main indicators that were identified highlight the existence of an aging rural population that shows an evident decoupling with traditional agricultural socioeconomics. The loss of rurality is related to the increase in urban land area and the development of the service sector (Figures 5 and 6). The social-ecological change from rural to urban systems causes rural decoupling and its corresponding ecological, social, and economic consequences [15,19,25].

Protection efforts through the establishment of the PA network have not prevented the processes of degradation of the cultural landscape, the decline of HNVF, and the loss of rurality that prevail in the area, mainly due to both the abandonment of traditional land uses such as afforestation and urban expansion [20,61,62]. On the contrary, in 2013, three years after the last analysed period, a large part of the study area was declared a National Park, the highest Spanish protection category (Figure 1). The conditions of the territory at the time of the establishment of the National Park were typical of a cultural landscape immersed in a process of transformation, rewilding, and rural marginalization. After the declaration of the National Park, it is foreseeable that the HNVF characteristics of the area will accentuate their degradation process as a consequence of the application of restrictions to local development and the practice of traditional land uses included in the normative schemes of this management category in Spain to promote land uses and activities more related to naturalness than to the protection of the cultural aspects of the territory [22,42].

## 5. Conclusions

In this paper, we quantify the evolution and current situation of the HNVF belonging to a historically cultural landscape that houses an extensive network of Pas that have been established for decades and that continue to be developed.

The obtained results highlight a significant loss of HNVF, mainly represented by traditional livestock systems (pastures and dehesas), and a marked increase in rewilding processes characterised by scrub–forest transition and the development of forest systems. The observed decrease of HNVF is linked to the disruption of the transmission of TEK and the decline of traditional land uses and practices, which may imply negative consequences both for the high biocultural diversity that these systems host and the cultural identity and the socioeconomics of these rural populations. Thus, the identified socioeconomic indicators reveal the decoupling between the rural population and traditional agricultural socioeconomics. The loss of rurality is mainly related to the transition from an agriculture-based economy to a service-based economy and urban development.

This social–ecological dynamic has been identified both inside of and outside of the boundaries of the PAs, so the transformation of the rural cultural landscape and the abandonment and loss of their HNVF seems to be a generalized process independent of the status or protection category of the territory.

The used method is easily replicable and useful in social-ecological land planning and in the design and implementation of effective management plans for the conservation of rural cultural landscapes as well as in testing the effectiveness of PAs. The design inside of

and outside of PAs has proven to be successful in achieving the proposed objectives. Thus, the degradation of the rural landscape and the vulnerability of the HNVF inside the limits of the established PAs reveal the ineffectiveness of their conservation plans, which do not favour the maintenance of traditional rural systems.

In this degraded cultural landscape, a National Park has recently been declared. In Spain, this land protection category is aimed at a type of conservation based on the restriction of human intervention in the environment. The establishment of the National Park has prioritized rewilding processes through land abandonment and the protection of forest landscapes over the conservation of traditional grassland systems of high natural value, which have hardly been considered.

Since rural cultural landscapes and their associated HNVF largely depend on the assesments and decisions of society, our results raise some relevant questions: (i) Are we designing and applying the appropriate management strategies to guarantee the sustainable future of cultural landscapes?; (ii) Are the current regulatory and normative frameworks for PAs really effective in conserving the cultural values and biodiversity of the landscape?

**Author Contributions:** Conceptualization, methodology, software, analysis, data curation, writing—original draft preparation, writing—review and editing, and supervision: M.F.S., C.A.-S. and P.S.-M.; funding acquisition: M.F.S. and C.A.-S. All authors have read and agreed to the published version of the manuscript.

**Funding:** This research was funded by the project LABPA-CM: Contemporary Criteria, Methods and Techniques for Landscape Knowledge and Conservation (H2019/HUM-5692), funded by the European Social Fund and the Madrid Regional Government.

**Institutional Review Board Statement:** Not applicable.

**Informed Consent Statement:** Not applicable.

**Data Availability Statement:** Not applicable.

**Acknowledgments:** The authors thank the Ministry of Science and Innovation and the State Research Agency as well as the postdoctoral support provided through the Juan de la Cierva training program (FECI2019-040562-I/AEI/10.13039/50110001103).

**Conflicts of Interest:** The authors declare no conflict of interest.

## Appendix A

**Table A1.** Municipalities of the study area inside of and outside of the protected area network: "Cuenca Alta de Manzanares Regional Park", "Cumbre, Circo and Lagunas de Peñalara Natural Park", and "Sierra de Guadarrama National Park".

| Municipalities | Municipality Code | Municipal Area within Regional Park Boundaries (%) | Municipal Area within Natural Park Boundaries (%) | Municipal Area within SG National Park Boundaries (%) | Municipal Area within Peripheral Protection Zone of National Park (%) |
|---|---|---|---|---|---|
| Alameda del Valle | 1 | 0.0 | 0.0 | 25.5 | 72.4 |
| Alcobendas * | 2 | 10.2 | - | 0.0 | 0.0 |
| Algete * | 3 | 0.0 | - | 0.0 | 0.0 |
| Alpedrete * | 4 | 0.0 | - | 0.0 | 0.0 |
| Becerril de la Sierra | 5 | 57.6 | - | 20.9 | 16.2 |
| El Boalo | 6 | 75.3 | - | 16.8 | 26.0 |
| Bustarviejo * | 7 | 0.0 | - | 0.0 | 0.0 |
| Canencia de la Sierra | 8 | 0.0 | - | 0.0 | 82.6 |
| Cercedilla | 9 | 62.6 | - | 28.1 | 52.1 |
| Colmenar Viejo | 10 | 29.6 | - | 0.0 | 0.0 |
| Collado Mediano * | 11 | 0.0 | - | 0.0 | 0.0 |
| Collado Villalba | 12 | 38.1 | - | 0.0 | 0.0 |
| Galapagar * | 13 | 7.3 | - | 0.0 | 0.0 |
| Garganta de los Montes * | 14 | 0.0 | - | 0.0 | 0.0 |
| Gargantilla del Lozoya * | 15 | 0.0 | - | 0.0 | 0.0 |
| Guadarrama * | 16 | 0.0 | - | 0.0 | 1.3 |

**Table A1.** *Cont.*

| Municipalities | Municipality Code | Municipal Area within Regional Park Boundaries (%) | Municipal Area within Natural Park Boundaries (%) | Municipal Area within SG National Park Boundaries (%) | Municipal Area within Peripheral Protection Zone of National Park (%) |
|---|---|---|---|---|---|
| Hoyo de Manzanares | 17 | 100.0 | - | 0.0 | 0.0 |
| Lozoya | 18 | 0.0 | - | 22.3 | 74.1 |
| Majadahonda * | 19 | 0.0 | - | 0.0 | 0.0 |
| Manzanares el Real | 20 | 98.9 | - | 55.0 | 11.3 |
| Miraflores de la Sierra | 21 | 52.7 | - | 7.3 | 48.2 |
| El Molar * | 22 | 0.0 | - | 0.0 | 0.0 |
| Los Molinos * | 23 | 0.1 | - | 0.0 | 44.1 |
| Moralzarzal | 24 | 64.5 | - | 0.0 | 0.0 |
| Navacerrada | 25 | 64.1 | - | 32.9 | 36.6 |
| Navalafuente * | 26 | 0.0 | - | 0.0 | 0.0 |
| Navarredonda y San Mamés | 27 | 0.0 | - | 3.6 | 27.3 |
| Pedrezuela * | 28 | 0.0 | - | 0.0 | 0.0 |
| Pinilla del Valle | 29 | 0.0 | - | 27.6 | 71.4 |
| Rascafría | 30 | 0.0 | 100.0 | 52.8 | 43.7 |
| Las Rozas de Madrid | 31 | 37.0 | - | 0.0 | 0.0 |
| San Agustín de Guadalix * | 32 | 0.1 | - | 0.0 | 0.0 |
| San Sebastián de los Reyes * | 33 | 14.9 | - | 0.0 | 0.0 |
| Soto del Real | 34 | 43.9 | - | 3.6 | 27.3 |
| Torrelodones | 35 | 58.3 | - | 0.0 | 0.0 |
| Tres Cantos | 36 | 100.0 | - | 0.0 | 0.0 |
| El Vellón * | 37 | 0.0 | - | 0.0 | 0.0 |
| Villanueva del Lozoya * | 38 | 0.0 | - | 0.0 | 0.0 |

The municipality code used in analyses and graphs and the municipal area within the boundaries of the parks are indicated. An asterisk indicates municipalities with less than 25% of their area within the park network boundaries, or municipalities that are not included at all within the park boundaries.

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
