# Peer review of "High Nature Value Farming Systems and Protected Areas: Conservation Opportunities or Land Abandonment? A Study Case in the Madrid Region (Spain)"

_land, doi:10.3390/land10070721_

Round 1

Reviewer 1 Report

The innovativeness of the research is limited, but the article is well organized and correct. I would like to point out an abuse of acronyms. Some acronisms are used few times in the text and therefore useless. Sometimes you get confused yourself for example HVN in the line 165 or HNVF at line 185 etc. I suggest to reduce acronyms to five or six in the whole text and to introduce a table of acronyms if you exceed 6. Check line 74 (two times rewilding). Check the definition of Mediterranean climate in agreement to Köppen.  Check in table 1 "%". Correct 200 in line 203. Check "x HNVF" in line 227. TEK has not been declared. Abstract may be improved changing the last 6 lines with the main results. 

Author Response

Thank you for your kind appreciation of the paper and your suggestions for improving it. We have made the indicated changes.

  1. Regarding the acronyms, following your suggestion we have left six in the whole text. Thus: i) We have removed the acronym corresponding to Social-Ecological Systems (SESs); ii) We have corrected the acronym of high nature valued farming systems, now it is HNVF always; iii) We have removed the acronyms for Regional Park (CAMRP), Natural Park (PNP) and National Park (SGNP).
  2. We have eliminated (%) from Table 1. The unit of measure of the land use variables appears in the Table legend.
  3. Line 74 (two times rewilding). Thanks, sorry it was a mistake. We have corrected it.
  4. Thanks, we have added the definition of Mediterranean climate according to the Köppen classification (lines 121-122).
  5. In the first paragraph of the introduction section, reference is made for the first time to Traditional Ecological Knowledge (TEK), defined by Berkes et al. (2000).
  6. Following your suggestion, we have modified the last part of the abstract highlighting the most relevant results.

Reviewer 2 Report

The paper is interesting and altogether well written. Section 3.3, however, was difficult for me to understand. The role/influence of the national park should be presented more clearly. Figure 7 should be described better. For instance, how was the shaded area determined? Also, please describe better how it relates to the other data points in the figure and what you conclude from this. This section is quite short compared with the other ones and I think there is enough space of some more detail – especially given that the national park is one of the core issues in the paper.

Minor points:

Probably a typo: In lines 31 and 32 you abbreviate social-ecological system as SSE, but it should be SES – as in line 63.

Lines 73-74: I understand that you describe two “contrary” effects of rewilding, but denoting both of them with the same word, rewilding, is a bit confusing. Better use two different words, or explicitly state that these are two consequences of rewilding.

Author Response

Thank you for your review of the paper and your suggestions to improve it. Below we indicate to you the changes we have made to the manuscript.

Section 3.3. Thanks, following your suggestions we have rewritten this section (lines 320-345 of the new manuscript). We have detailed the meaning of the area shaded in blue in Figure 7 and the social-ecological implications of the selection of the municipalities that will be part of the future National Park. Now this section is clearer.

Minor Points

  • Thanks, it was a mistake. In any case, we have removed the acronym corresponding to Social-Ecological Systems (SESs) in the whole text.
  • Thanks, it was another mistake. On line 74 it appears twice rewilding. We have corrected it in the new manuscript: (lines 72-75) These processes of rewilding (returning ecosystems to a higher naturalness, seeking wildlife comeback without human intervention), cause the disappearance of HNVF, fostering spatial homogeneity and the degradation of the cultural landscape.